# Selected Aspects of Precision Grinding Processes Optimization

**DOI:** 10.3390/ma17030607

**Published:** 2024-01-26

**Authors:** Wojciech Kacalak, Dariusz Lipiński, Filip Szafraniec

**Affiliations:** Faculty of Mechanical Engineering, Koszalin University of Technology, Racławicka 15, 75-620 Koszalin, Poland

**Keywords:** grinding, optimization, durability, tool life, surface quality, topography, dimensional accuracy

## Abstract

The paper describes selected aspects of the optimization of grinding processes, taking into account the characteristic probabilistic features of this process. Characteristic features of the grinding process that influence the significant dispersion of the quantities used in the optimization process to define goals and limitations are indicated. Attention was paid to the reasons for uncertainty in the use of research results, imperfections in information extraction procedures and the limited amount of data in the use of simulation and regression models in optimization procedures. The issue of determining the durability of abrasive tools in grinding process optimization procedures was analyzed. Methodologies for defining tool life are specified, taking into account the dispersion of the values of controlled process parameters. The effects of interference were taken into account in the relationships describing grinding efficiency and costs. The benefits of optimization taking into account the probabilistic nature of the process were determined.

## 1. Introduction

Grinding processes, often used as the final operation in the manufacturing process. These processes have a decisive impact on the accuracy of the shape and dimensions of the manufactured elements [1], determining the surface topography and the condition of the surface layer, and influencing the functionality of the mating surfaces [2] and the durability of manufactured elements [3]. Grinding processes also have significant economic importance, often accounting for up to 20 to 25% of the total costs of the manufacturing process [4]. For these reasons, optimization of the grinding process is an issue of key importance for the final results of the entire manufacturing process.

The correct definition of goals and criteria for optimizing grinding processes requires knowledge of the nature and variability of phenomena occurring in this process, allowing the determination of factors that have a significant impact on the results of the grinding process. These factors belong to four main groups:Factors depending on grinding parameters and conditions—resulting directly from the process parameters and tools used; these factors affect the thermal and mechanical deformations of the machining system and its vibrations; by changing processing parameters, it is possible to control the effects caused by these factors.Factors independent of machining parameters—resulting from the characteristic properties of the workpiece, e.g., stresses occurring in the semi-finished product; also included are factors with a limited degree of control over the effects of their interactions.Time-varying factors—resulting from factors characterizing the type and properties of the machine tool and the machining tools used.Disturbances—unmeasurable and uncontrollable quantities; it is possible to limit the unfavorable impact of disruptions on the technological process by building systems that compensate for the effects of their impacts.

The basic criterion in the optimization of grinding processes is the resulting accuracy of the shape and dimensions of the machined element [5] and the quality of its surface [6,7]. These factors are influenced by the appropriate selection of grinding parameters and conditions, which should also take into account the material properties, the material separation mechanism, and the magnitude of thermal and mechanical interactions in the contact zone of the tool with the workpiece. These factors may significantly influence differences in the efficiency and durability of manufactured elements [8,9].

The type and properties of the tool influence the magnitude of mechanical and thermal impacts in the grinding zone. It should be noted that as a result of wear processes, the condition of the active surface of the grinding wheel changes, also contributing to the variability of the results of the grinding process [2,10,11]. The properties of the grinding wheel (type and size of abrasive grains, type of binder, structure of the grinding wheel) affect important parameters determining the magnitude of interactions in the grinding zone [12]: cross-sections of the cut layers, contact surface of the grain with the workpiece, type and size of wear processes of abrasive grains and binder. The type of abrasive grains and the composition of the binder fundamentally affect the properties of the grinding wheel and determine the wear mechanism and wear rate, as well as the thermal conditions under which the workpiece and the grinding wheel interact [13].

In the optimization of the grinding process, it is impossible to ignore important factors related to the material separation process, which determines the resulting surface roughness, which is the main goal of optimization and an important final requirement of the manufactured parts [14,15]. The results of research including, among others, Refs. [16,17] indicate that when optimizing the efficiency of material removal, the influence of grain shape, the state of kinematic interactions and friction in the cutting zone should be taken into account. The shape of the abrasive grains also determines the state of material displacement around the abrasive grain tip and the chip formation mechanism, and consequently the value of grinding forces [18,19]. The value of grinding forces is significantly influenced by the feed speed and cutting depth, while the cutting speed and rake angle of the cutting edges significantly affect the temperature in the cutting zone [20]. Moreover, the maximum temperature values occur on the front surface of the cutting edge, and the temperature increase in the machining zone occurs after the initiation of the chip formation process [21].

High temperatures in the grinding zone may lead to the growth of material grains, phase transformations and, consequently, changes in mechanical interactions in the near-surface zones, i.e., the formation of burns and cracks [22,23]. Moreover, changes in the strain rate in the cutting zone [24], the cross-sections of the removed layers [25] and the properties of the workpiece [26] affect the mechanisms of initiation of the material removal process and the chip morphology. Moreover, the material properties in the contact zone of the abrasive grain with the processed material, due to changes in the near-surface microstructure, may differ significantly from the properties of the material [27,28].

In the optimization of grinding processes, an important factor influencing the final effects of the grinding process are the properties of the abrasive tool in contact with the workpiece. In precision grinding processes, the probabilistic features of the abrasive tool and the stochastic features of the process as well as the significant variety of machining procedures result in a significant level of uncertainty in predicting machining results. It is also difficult to optimize technological operations due to the diversity of processes with high dynamics of local phenomena and their significant variability over time. Most random factors in grinding processes result from the construction of grinding wheels. This group of factors includes random sizes and shapes of abrasive grains and their random arrangement on the tool surface. This is compounded by stochastic mechanical, physical and chemical wear processes of abrasive grains.

This paper describes selected aspects of optimization of grinding processes, taking into account the characteristic features of this process. The main reasons for the significant dispersion of values characterizing the course and results of grinding were indicated. Attention was paid to the causes of uncertainty in the use of research results, imperfections in information extraction procedures and the limited amount of data. The effects of interference were taken into account in the relationships describing grinding efficiency and costs. The benefits of optimization taking into account the probabilistic nature of the process were determined.

## 2. Results of Grinding Process Analyses Are Important for Optimization Procedures

### 2.1. Process Features

The cutting depth of the grains is usually smaller than a few micrometers, and therefore, it is smaller than the rounding radii of the grain tips. The length of the contact path of the grains with the machining surface is from 1 to 1000 μm. The grain depth along this path is variable and depends on the surface topography in the contact zone and on the variability of the grain path. The local values of specific processing energy, ranging from 20 to 500 J/mm^3^, are higher in zones with small grain depth [29]. This shows that the grinding energy depends not only on the machining efficiency and the average cross-sections of the cut layers, but also on the form of the distribution of cross-sections of the cut layers at a given moment. Grinding energy is a factor that determines not only the properties of the surface layer, but also tool life and machining efficiency [4].

The variability of deformations in the zone of influence of individual grains causes discontinuity in the formation of microchips. The frequency of microchip formation may reach several MHz, and the thickness of microchip plates (Figure 1) may be less than 0.1 micrometer.

SEM images in Figure 1 show the results of the grinding process of elements made of X153CrMoV12 steel (NC11LV 1.2379) with a hardness of 63 HRC with a grinding wheel 1-250 × 25 × 76.2-99A100K7VTE10-35 with a peripheral speed of 35 m/s, longitudinal feed 25 m/min and transverse feed 0.5 mm. It is seen that high local temperatures cause tensile stresses in the surface layer and periodic formation of hollow spherical microchips (Figure 1b).

The importance of the results of the analysis of the impact of temperature on the grinding process in the selection of grinding parameters was highlighted in [30]. The problems of validating thermal models of the contact zone are described and the results from the use of an infrared measurement system are given. The analyses of thermal transformations in the grinding zone were carried out in [31]. The authors developed a heat flux model taking into account the process parameters and grinding wheel characteristics.

Modeling the grinding process and creating the basis for predicting process results has long been an effective method of creating optimization models [4]. Many different models have been developed in previous research, especially grinding forces [32,33]. This issue was developed in many other works, including [12,34].

The development of modeling of abrasive machining processes consisted in improving empirical relationships, also using artificial neural networks [35]. In subsequent works, the possibilities of creating models using multidimensional analyses were used [36]. In other works, energy aspects for microchip formation and friction forces in the cutting zone have been described [37,38].

The accuracy of modeling grinding forces was improved as a result of work that took into account the properties of the grinding wheel topography [39]. Stochastic modeling has contributed to the development of modeling methods [40]. This made it possible to take into account the distribution of grains and determine the sum of forces from individual grains. Taking into account grain trajectories [41] extended the applications of the developed models.

Methods are also used to support the abrasive machining process using additional kinematic and dynamic effects [42,43]. The use of ultrasonic in the grinding process of 42CrMo4 material without cooling showed a beneficial effect of this type of machining on reducing surface roughness and reducing the grinding force by 60% [44].

Experimental research, modeling and simulation, and optimization of grinding processes are among the tasks that require the use of incomplete, uncertain, and, in the case of fuzzy reasoning, also inaccurate information. The number of events in the form of grain contacts with the processed surface can range from 10^6^ to 10^10^ per second. For an exemplary process, the following criteria are used: grinding wheel diameter *D* = 400 mm, grinding wheel height *H* = 50 mm, grain size *a_z_* = 120 µm; there are about 10^6^ abrasive grains on the surface of the grinding wheel, of which about 50,000 can be active. For a speed of 35 m/s, the number of grain contacts with the surface in each second of the process is over 2.2 × 10^6^. To this should be added the high frequency (0.2–5 MHz) of shearing of the micrometric layers forming the chips.

The ratio of the standard deviation σ and the average value m, for many quantities, such as the momentary value of force, local variations in the values of roughness parameters, local values of dimensional or shape deviations, can reach values close to 1.

The main reasons for the significant dispersion of these magnitude values are as follows:Variation of machining conditions in subsequent operations, resulting from the variability of the machining allowance and the movement of the machining zone along a specific path, which changes the deformation of the system.Changes in the condition of the working surface of the grinding wheel, the dynamics of which are subject to fluctuations and variability during the tool life.Progressive form wear of the grinding wheel, causing deviations in dimensions and shape and enlarging the area of the grinding zone.Fluctuations in the flow of machining fluid through the machining zone.Vibration of the tool and workpiece.

### 2.2. Optimization Criteria Regarding the Accuracy of Machining and the Geometric Structure of the Surface

The possible accuracy of machining is assessed at the stage of creating the structure design. In this phase, it takes into account various limitations occurring in the production and quality control stages. Decisions regarding the required machining accuracy are subject to many constraints. The planned machining accuracy is therefore lower than the highest accuracy that can be achieved using a given machining method.

The limits of the accuracy achieved in shaping the geometry of designed elements are shifting as a result of the improvement and dissemination of precise machining methods. Decisions on the selection of machining accuracy in grinding processes usually depend on technological and economic factors and the characteristics of the material being processed, the machining method used and the properties of the grinding wheel.

Technological factors affecting the accuracy of manufactured elements mainly include limitations resulting from the selection of technological devices, tools, processing parameters, and supervision and control methods. Additionally, limitations resulting from standardization and the applied principles of element interchangeability can be pointed out.

The group of economic factors includes limitations regarding acceptable costs, limitations resulting from the acceptable price of the product as a whole, depending on the market conditions for specific products, as well as operational aspects, reliability and features resulting from the preferences of customers.

Factors dependent on the selection of grinding parameters and conditions include limitations resulting from the features of the process, such as the number of cutting blades shaping a specific part of the surface, variability of the grinding wheel surface condition as a result of progressive wear, formation of side ridges around machining marks, deformations in the technological system, and kinematic inaccuracy machining.

The basis for decisions regarding accuracy in the designed grinding processes may be the sequence of research procedures presented in Figure 2. Decisions regarding the accuracy of grinding operations usually result from a forecast of the results of future process implementations created on the basis of the knowledge of decision makers or data from a specific process implementation. The form of the distribution of values of the considered feature is influenced by the mechanism of accumulation of many causes of deviations.

In processes considered stationary, we can distinguish additive, multiplicative, vector additive and other hybrid mechanisms of accumulation of factors causing deviations in dimensions and shape. In processes in which the average value of the deviation and (or) the dispersion of its value changes during the implementation of many operations in the analyzed process, the resulting distribution depends on the scope of these changes.

An example is the results of the momentary dispersion of flatness deviations and the total distribution of flatness deviations (Figure 3) of small ceramic elements processed in an automatic cycle using a special AR7 grinder (Koszalin University of Technology, Koszalin, Poland) (Figure 4) [29].

Precision grinding processes are usually non-stationary processes due to changes in the topography of the grinding wheel and differences in the grain cutting depth and workpiece properties. The total distribution of deviations from a certain period of production is then of a different type than the distributions from shorter periods and depends on the starting point and time of data collection.

A deterministic analysis of the causes of deviations in the position of the tool surface relative to the machined surface allows for the determination of the most important causes of inaccuracies. However, it does not allow for the assessment of the form of dispersion of deviation values, which limits the prediction of machining results. This is especially visible in the case of grinding elements with complex shapes, such as screw surfaces, with involute contours, or very accurate machining of elements with defined, within tolerance dimensions, number of cases in specific ranges.

Deviations in the dimensions or shape of grinded elements may be the result of many factors with a very small impact, none of which dominates over the others. Then, the analysis of the impacts and accumulation of many outflows allows us to determine the form of deviation distributions and their limit values.

In grinding processes, deviations in dimensions and shape are most often the result of vector summation of deviations in relative displacements of the tool and the object. There is no scalar summation mechanism. It should be emphasized that the resulting distribution is then different from the normal distribution. Even if the normal distribution approximates the values in the zone of average values quite well, the agreement with the data decreases in the ranges of minimum and maximum values. This may lead to incorrect decisions regarding the frequency of extreme values.

It should be emphasized that the lognormal distribution (corresponding to the multiplication mechanism) often creates a composition with other distributions, for example, the chi distribution or the Rayleigh distribution. Then, comparing the accuracy of extreme zones with values with low occurrence frequencies is a condition for correct diagnostic decisions. The multiplication mechanism is typical for gradual material removal processes in surface grinding with transverse feed. In the case of many deviations in the position of elements of a technological system with different directions, the dispersion of the resulting deviation value, as the geometric sum of component deviations, can be approximated by the Rayleigh distribution.

When defining and controlling the deviations of ground elements, in many cases, the maximum value of a specific deviation is determined. Then, in a set of many elements, the form of the maximum value distribution is assessed. It can then be assumed that the probability of specific values decreases exponentially as these values increase.

Analyzing topography, assessing stereometric properties and predicting the operational characteristics of technical surfaces is an increasingly important task for designers of modern products. The increase in accuracy requirements and the pursuit of higher durability of elements resulted in the development of methods for measuring the geometric structure of surfaces. This has greatly expanded the scope of analysis required and the domain of choices that must occur in the structural design process.

An important task is the selection of stereometric parameters of technical surfaces that create a complementary set and demonstrate classification effectiveness and technologically useful interpretation of assessments (Figure 5).

The surfaces of the designed elements differ, among other ways, in the shape and distribution of irregularities, features of specific motifs, spectral features, the degree of surface development, the occurrence of porosity and peaks, the sharpness and height of elevations. The statistical features of the surface after grinding vary with respect to recesses and elevations. Surface areas in the form of elevations have a higher surface development and a higher density of peaks. The surfaces of the recesses are characterized by less surface development and less variability of local profiles.

Research [46] on the contact area of two surfaces shaped in the process of fine grinding showed that the number of contact areas for perpendicular associations of machining traces is small for low pressure forces and increases significantly for higher pressure forces. For parallel associations of machining marks, the number of contact areas is quite large for low pressure forces, but increases to a lesser extent for higher pressure forces.

The measure of surface contact properties can be statistical evaluations of the product of the absolute value of the gradient and the height of the location for a set of points in areas of the surface above a set level. The higher the value of this product, the smaller the sum of the contact areas of surface irregularities. The analyses carried out [47] show that smoothing the surface, resulting in a reduction in the amplitude of high-gradient peaks, and perpendicular association of machining marks on two contacting ground surfaces, allows for a significant increase in contact stiffness.

In [45], it was shown that many parameters for assessing the geometric features of surfaces do not differentiate the assessments in the analyzed set of surfaces. Parameters relating to the geometric features of motifs created by the highest vertices of the surface have a high ability to differentiate. The main problem in striving to increase accuracy and shape favorable properties of the surface layer in grinding processes with small allowances is the increase in specific energy with decreasing cross-sections of layers cut with individual abrasive grains.

The beneficial features of using a small grinding depth result from the reduction in the total energy of the process (despite the increase in specific energy), which helps to obtain more favorable features of the surface layer. Reducing the grinding allowance reduces the density of heat fluxes and shortens the time of local thermal exposure. This is the result of a reduction in the length of the contact zone between the tool and the workpiece. Using small grinding depths reduces forces, which is important when machining items with small dimensions and low stiffness.

When grinding with small allowances, the share of grains that shape the surface is reduced. This can be prevented by reducing the dispersion of the radii on which the abrasive grain blades act, using small-sized grains, reducing the distance between active grains, using abrasive aggregates or binders with locally reduced susceptibility.

However, it should be taken into account that the use of small grinding depths increases the movement of material in the form of side ridges (increases the share of plastic deformations). Large widths of layers cut, for example, by abrasive aggregates, limit lateral movements of the material. Another advantageous procedure is the use of high values of longitudinal feed speed, which increases the length of the cutting trace and reduces the share of grain entry and exit zones from the contact zone. Then, the transverse feed should be small to obtain a large number of micro-cutting marks per unit area. While maintaining a constant value of the product of depth, longitudinal and transverse feed (i.e., constant volumetric efficiency), it is beneficial to increase the longitudinal feed and reduce the transverse feed.

### 2.3. The Importance of Abrasive Tool Life in Procedures for Optimizing Grinding Operations

Optimization of grinding parameters can be carried out experimentally or using a mathematical model. The first method is used in cases where it is intended to determine the optimal process parameters only once for set conditions. The cognitive significance of this method is small, which is why optimization results using a model are more common in the literature.

In solving problems of optimizing grinding operations using models, due to the high degree of randomness, the first task is to create a model that is most consistent with the mechanism of variability of the analyzed input and output quantities. This is of great importance for process diagnostics and forecasting their results.

The problem can be simplified by pre-determining a narrow region of input quantities, close to the expected extreme of the objective function. However, it is not advisable to adopt the empirically best model due to the accuracy of the approximation for the adopted criterion. Otherwise, it could happen that in subsequent implementations of the process, for a certain set of model forms, the highest accuracy is achieved for a different model form. This would mean that instead of a model consistent with the process features, the user would use a model describing a set of data, which will probably be different in other implementations.

Polynomial models of variability of specific features of the grinding process are sometimes encountered. In such models, even a small change in the value in the data set (statistically justified) may cause significant changes in the values of the model coefficients, and even changes in the sign of the values of these coefficients. Another mistake is to designate models in an additive form. Such models for values of the independent variables (some or all) equal to zero show values that often make no physical sense. Multiplicative models do not have this disadvantage. It is then advantageous to take into account that the values of many features characterizing grinding processes are asymptotically limited because they can only change to a certain extent. This results in the usefulness of exponential functions with decreasing value of the first derivative.

In grinding processes, many parameters can be determined that could constitute an optimization criterion. These include, for example, process efficiency, operation costs, accuracy parameters, surface geometric structure parameters or surface layer properties. Constraints can be placed on some of these features, while others can remain optimization criteria.

An effective methodology may require defining one synthetic criterion using the following operations:The potentially useful optimization criteria should be normalized, preferably using fuzzy inference and determining the function of belonging to linguistic categories, meaning the favorable value of a given criterion;The sensitivity of each criterion for the adopted decision-making area should be assessed;Select those criteria that are most sensitive to changes in process parameters and are not strongly correlated;Create a synthetic criterion from them, preferably as a geometric mean of the component criteria.

In the tasks of optimizing grinding processes, the progressive wear of the tool with the cyclically renewed state of the active surface is the most important factor because it affects the forces, process energy and temperature field in the machining zone as well as the shaping of the geometric structure of the surface and the properties of the surface layer. This causes variability in important process features, which are usually subject to specific constraints. These limitations are related to the assessment of the machining potential, which remains to be determined automatically or by the operator and which is necessary to renew the condition of the tool’s active surface.

## 3. Determining a Grinding Wheel Life

In grinding processes, conditions limiting changes are imposed on established variables characterizing the process *K_proc_*, such as grinding forces and power, temperature at measurement points, and on parameters characterizing the results of the operation *K_oper_*, such as selected parameters of the geometric structure of the surface, stresses in the surface layer. The importance of changes in monitored variables for predicting the time remaining to the end of the grinding wheel life usually varies. Most often, one of the variables, taking into account the level remaining to the permissible limit and the dynamics of its changes, will influence the decision to end the tool’s life. This will be a variable that meets the condition
(1)$$K_{s}=\min(\min(K_{proc}),\min(K_{oper}))$$
where *K_s_* is the variable whose level is closest to the limit of the permissible value. The determination of variables of decisive importance should be more frequent, the shorter the period remaining until the limiting conditions are fully met.

There will be different periods after which individual monitored variables could reach the limit values if the process were continued despite the limit value being exceeded by the variable that is dominant in a given operation. It follows that the durability period of the grinding wheel *T_s_*, in which all monitored process variables and machining results are within the limits of their permissible deterioration, is determined by the relationship:(2)$$T_{s}=\min[T_{K_{s}}(K_{s\;min})]$$

This means that the durability of the grinding wheel *T_s_* is equal to the sum of the grinding times after which at least one of the process variables or quantities describing the process results reaches its limit value.

The *T_Ks_* function contains random components that depend on many of the previously described probabilistic features of the process (Figure 6), and mainly on the variability of the machining allowance in subsequent operations, the varied impact of tool wear and the variability of cooling conditions on the controlled variables.

Reducing the unit grinding energy and temperature in the grinding zone is one of the most important factors determining the quality of manufactured elements in many processes. Lowering the temperature in the surface layer during grinding with a fixed volumetric efficiency of the process can be achieved by increasing the speed of the heat source, i.e., the longitudinal feed, and proportionally reducing the transverse feed. Even if the efficiency increases proportionally with the increase in the longitudinal feed speed, favorable thermal conditions can be ensured in the grinding zone (Figure 7).

Figure 7 shows the temperature field in the grinding process of constructional steel for different velocities of the heat source (longitudinal grinding feed) 0.005, 0.01, 0.02, 0.04, 0.08, 0.16, 0.32 and 0.64 m/s, for an initial temperature of 290 K, heat source efficiency depending on the feed speed and ranging from 20 to 130 W/mm^2^, for the density of the material 7820 kg/m^3^, specific heat 490 J/(kg∙K) and thermal conductivity 42 W/(m∙K). The length and width of the cooling zone were 40 and 20 mm, respectively. The displacement of the cooling zone relative to the center of the grinding zone in the direction opposite to the direction of the feed movement was 10 mm.

The simulation results indicate beneficial thermal effects resulting from increasing the longitudinal feed speed. To ensure the required surface roughness for a given surface machining efficiency, it is possible to reduce the value of the transverse feed.

In the case of grinding with transverse feed, the use of such cooling systems causes the leading edge of the grinding wheel, which is the most heavily loaded, to be cooled to a lesser extent than the peripheral surface zones of the grinding wheel, which are mainly involved in the surface sparking process. This is particularly important in the grinding processes of materials that are difficult to machine, have a low heat capacity coefficient, a low thermal conductivity coefficient, high ductility and a tendency to harden as a result of thermal–mechanical interactions.

## 4. Productive Efficiency of Grinding Operations

The production efficiency of the grinding operation *W* and its cost *K* depend on the durability of the grinding wheel and the grinding parameters. The efficiency can be expressed by the following relationship:(3)$$W=\{\frac{t_{pz}}{n}+\left[t_{p}+t_{g}\left(Q_{v}\right)\cdot \left(1+\frac{t_{k}}{T_{s}\left(Q_{v},W_{g}\right)}\right)\right]\cdot \left(1+k_{u}\right){\}}^{-1}$$
where *t_pz_* is the preparatory and finishing time, *n* is the number of items in the series, *t_p_* is the auxiliary time, *t_g_*(*Q_v_*) is the main time, which depends on the volumetric grinding efficiency *Q_v_*, *t_k_* is the dressing time of the active surface of the grinding wheel, *k_u_* is the supplementary time, and *T_s_*(*Q_v_*,*W_g_*) is the durability of the grinding wheel depending on the set of controlled variables *Q_v_* and the set of limiting conditions *W_g_*, imposed on changes in these variables during the machining operation.

In grinding operations where a strategy of grinding wheel dressing in advance is used, the life of the grinding wheel may be determined by variables other than those in the case of continuous monitoring of the grinding process. For example, Figure 8 shows that the durability of a grinding wheel renewed in advance is determined by changes in feature *c*1 (normalized grinding force value) and then the durability is less than 40 min than in the case when the durability of the grinding wheel is determined by the results of continuous monitoring. Then, the durability is determined by changes in the value of the variable *c*2 (normalized value of the Sp5 parameter of the height of surface irregularities), and the average tool life exceeds 60 min.

Normalization of the values of these parameters to the range <0, 1> allowed for visualization of the problem and comparison of data in a common range. In the normalization performed, 0 corresponds to the lowest value of a given feature in the analyzed process, and 1 corresponds to the highest value that could occur during the durability of the grinding wheel.

Durability distributions determined by taking into account each variable separately will, of course, differ in expected value and variance. The form of the distribution, as a result of the mainly multiplicative mechanism of accumulation of the influence of factors influencing changes in individual variables, will be similar to the log-normal distribution.

From the above observations, it follows that under machining conditions in which the condition of the tool and process variables are monitored, in subsequent periods of grinding wheel use, various limiting conditions may determine the tool life, but the average life will be at a level much higher than that corresponding to dressing the grinding wheel in advance.

In grinding processes carried out in series production conditions, if they are not adaptively controlled, a specified uniform durability period of the grinding wheel is assumed. It is assumed that for specific grinding parameters *x_i_*, the assumed durability period *T_s_*(*x_i_*) will be such that the limiting conditions imposed on the selected values *y_j_* will be met throughout this time. The quantities *y_j_*, on which limiting conditions are imposed, depend on the grinding parameters *x_i_*, the operating time of the grinding wheel *t* since the last shaping of its active surface, and the disturbances *z*, the number and effects of which are unknown. From limiting conditions
(4)$$y_{j}=f_{o}\left(x_{i},t,z\right)\geq \{\left(\leq \right)\}y_{i\;dop}$$
it follows that the times *t_j_*, after which the quantities *y_j_* will reach the limit values, will depend on *x_i_*, *y_idop_* and disturbances *z*. This means that in subsequent operations, the times *t_jwmin_* = min(*t_j_*) will be different.

This results in a higher complexity of process optimization procedures (Figure 9), in which an important role is played by the grinder operator or, in still rare cases of adaptive control, by a decision support system for the selection of parameters. A new way of interaction between operators and technical devices is being developed, with two-way voice communication described in numerous works [48,49].

## 5. Summary

In grinding processes, there is a significant dispersion of values characterizing the course and results of grinding. The distribution of the examined variables depends on the mechanism of accumulation of the effects of random disturbances. In most cases, there is a multiplicative mechanism or, in relation to geometric features, an additive vector mechanism.

The durability of abrasive tools is a feature that depends on changes in the values of features describing the properties of the tool, which change with grinding time. Tool life is the shortest period during which selected normalized values of analyzed process variables and results of machining operations do not exceed permissible limits.

These variables show systematic and random changes. Analyses of tool life in grinding processes limited to changes in the expected value lead to significant errors in the assessment of tool life for strategies with dressing of the grinding wheel active surface in advance.

In a typical situation, when several features of the process or machining results are used to assess the durability of the grinding wheel under supervised conditions, it should be taken into account that in subsequent durability periods, the time remaining to renew the tool surface may be determined by variables other than in the previous period. This contributes to the significant complexity of the decision-making process in these tasks.

Monitoring the condition of the tool significantly improves the use of the machining potential of the tools, which means lower tool costs, lower machining costs, as well as an increase in efficiency as a result of reducing the number of dressings of the grinding wheel active surface.

## Figures and Tables

**Figure 1 materials-17-00607-f001:**
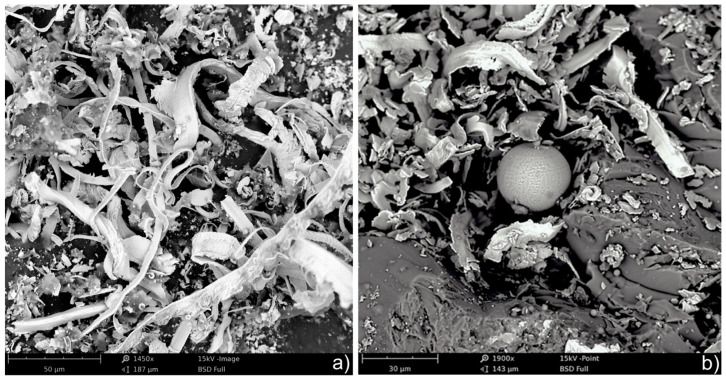
SEM images of microchips created in the grinding process of X153CrMoV12 steel (NC11LV) (**a**), microchips with a spherical hollow chip visible (**b**).

**Figure 2 materials-17-00607-f002:**
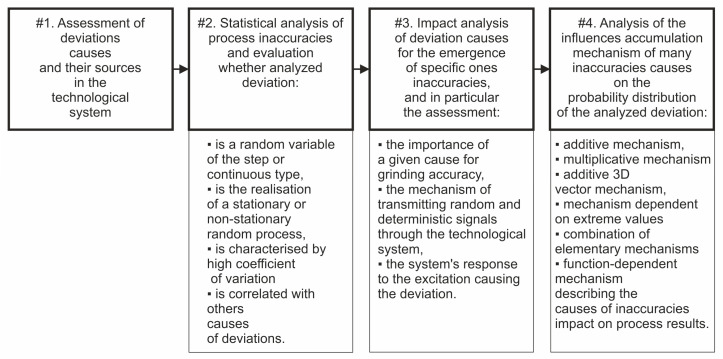
Diagram of the sequence of research procedures in the decision-making process regarding the expected machining accuracy.

**Figure 3 materials-17-00607-f003:**
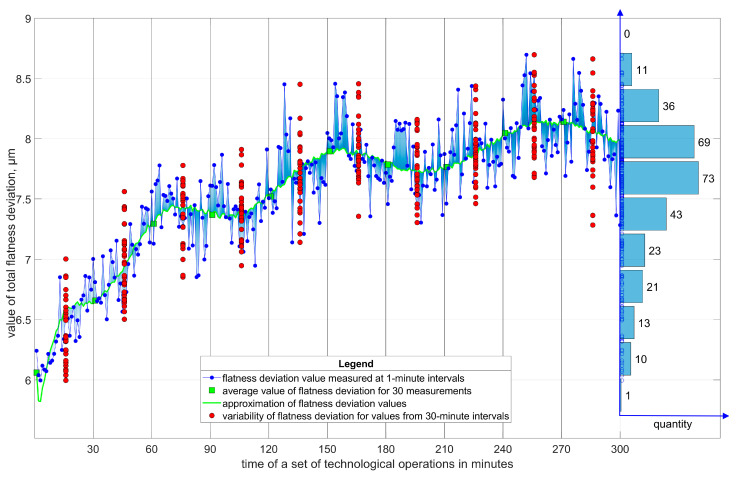
Momentary dispersion and total distribution of flatness deviation of ground small ceramic elements in the automated grinding process using the constructed AR7 machining system [29].

**Figure 4 materials-17-00607-f004:**
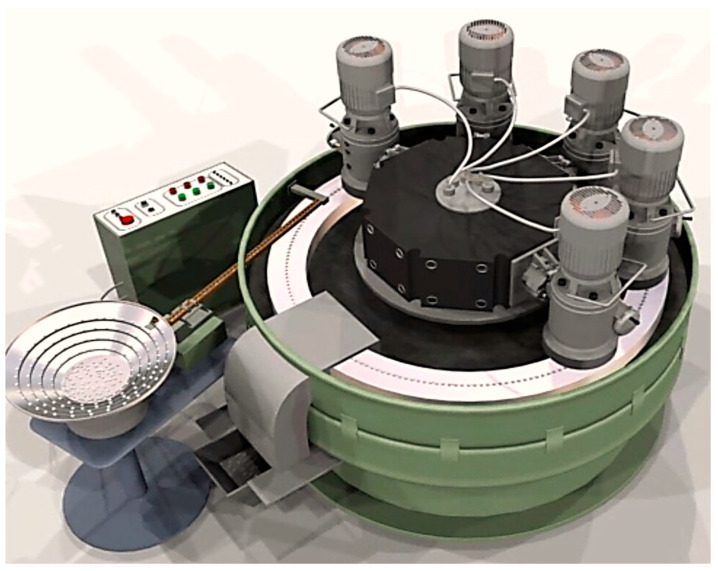
Grinder for automated grinding of small flat ceramic elements [29].

**Figure 5 materials-17-00607-f005:**
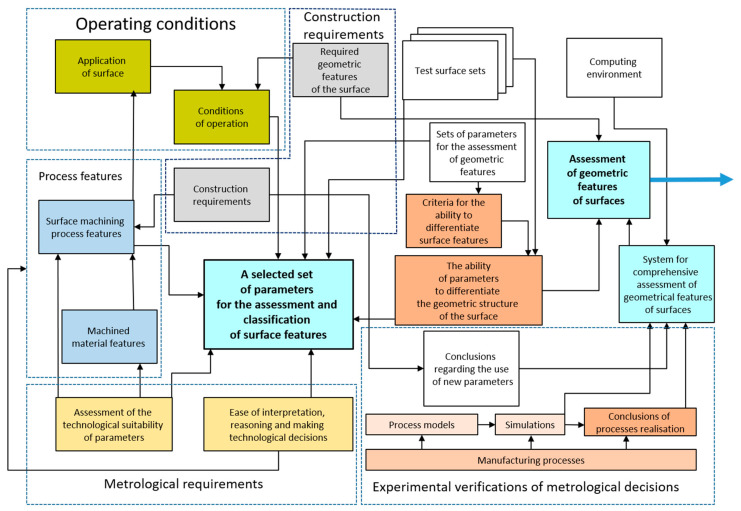
Scheme of the system for comprehensive assessment of geometric features of surfaces in the aspect of decisions in the process of designing technical elements [45].

**Figure 6 materials-17-00607-f006:**
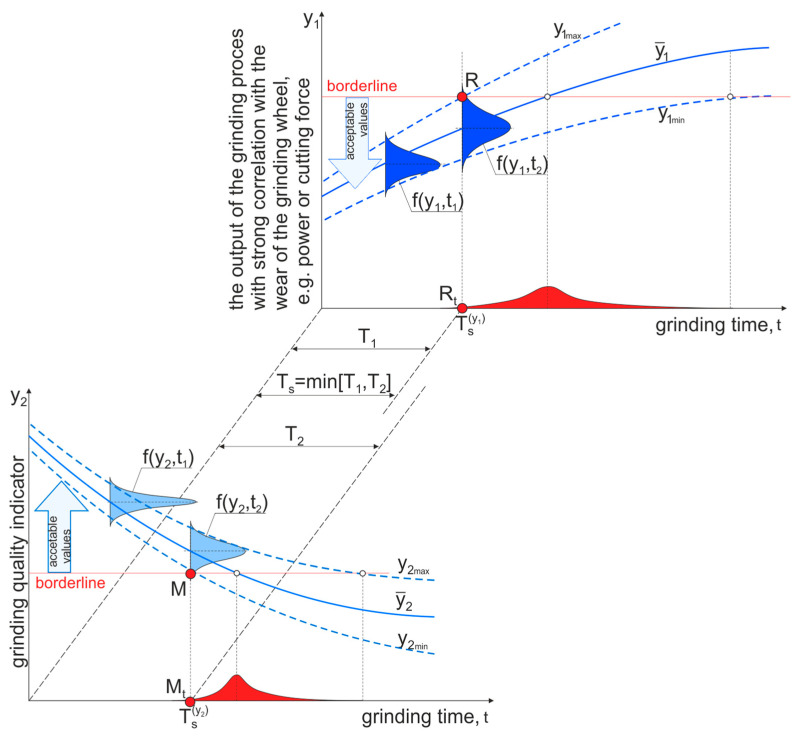
Scheme for illustrating the definition of the tool life taking into account the dispersion of the values of the controlled variables, on which conditions have been imposed that limit their permissible changes.

**Figure 7 materials-17-00607-f007:**
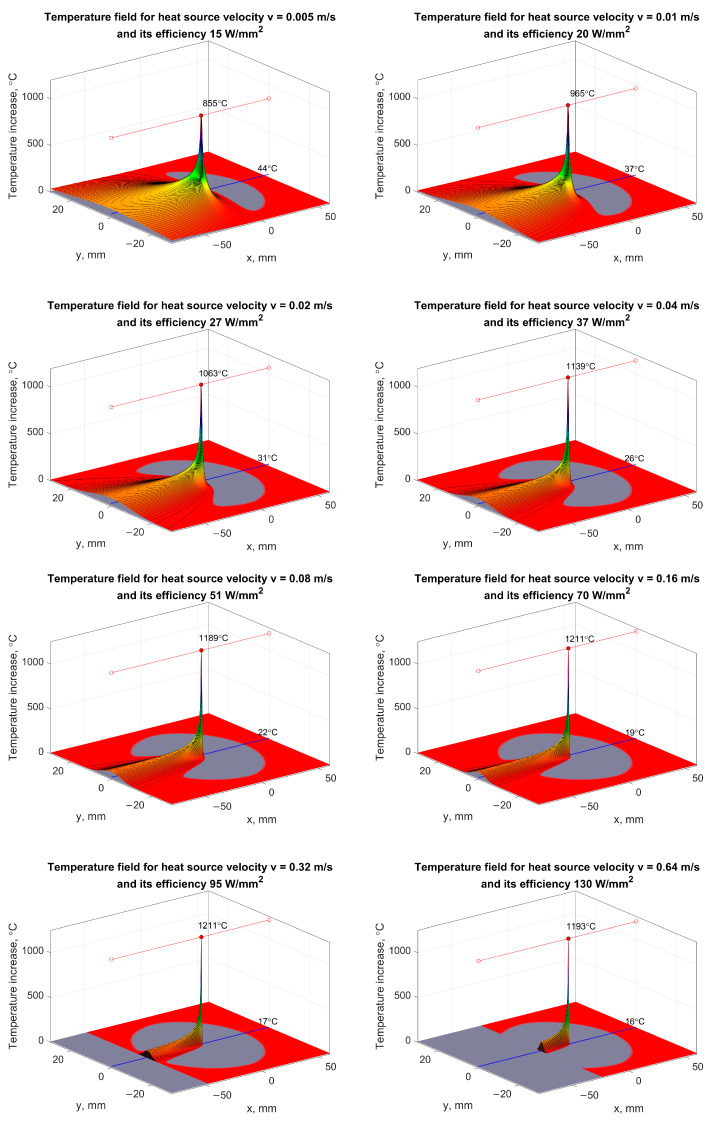
Illustration of the influence of the longitudinal feed speed on the temperature field around the grinding zone with cooling shifted back by the width of the grinding zone.

**Figure 8 materials-17-00607-f008:**
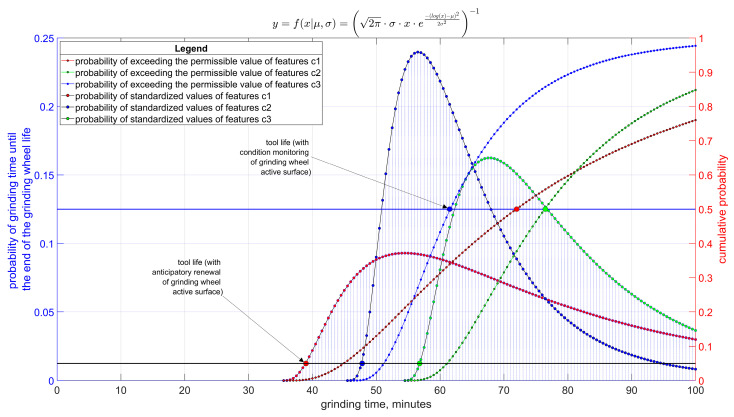
Distributions of the values of variables taken into account in the assessment of the time to the end of the grinding wheel life based on the control of various process variables for two strategies: dressing of the tool surface in advance, and supervision of the process state, where the following are defined: *c*1—normalized value of the grinding force; *c*2—normalized value of the Sp5 roughness parameter; *c*3—normalized value of the diameter deviation of the ground cylindrical surface.

**Figure 9 materials-17-00607-f009:**
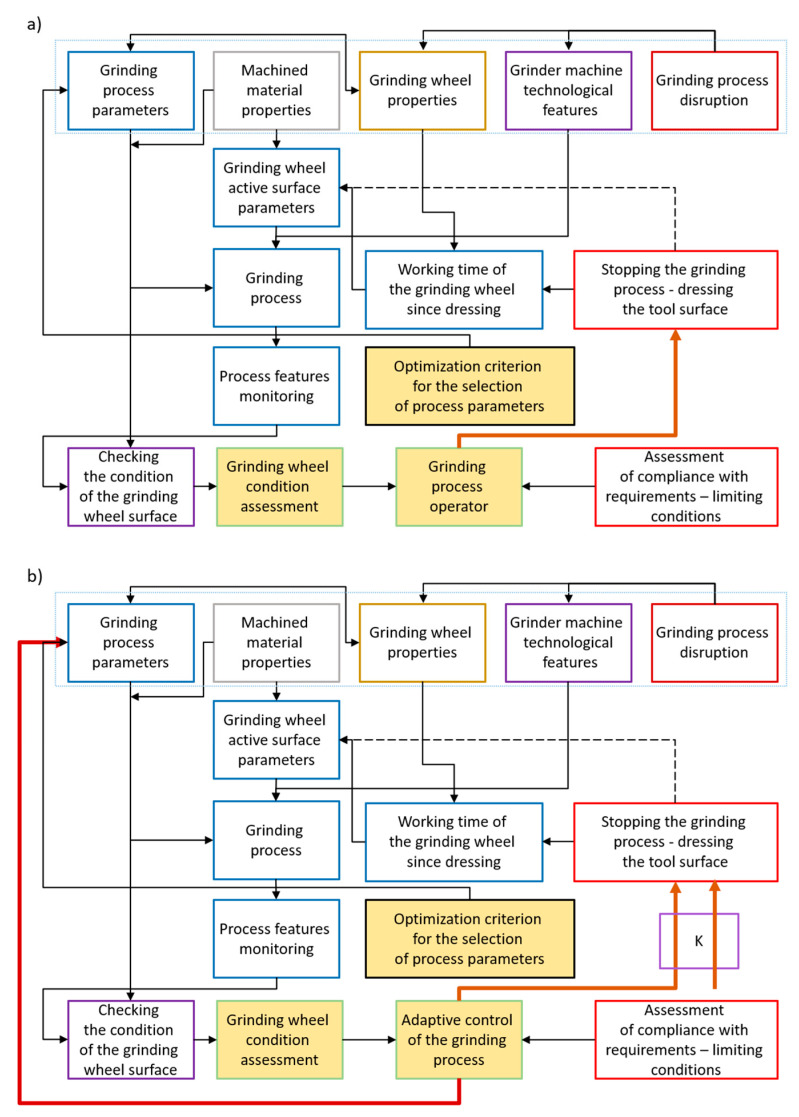
Process optimization procedures under operator supervision (**a**) and under adaptive process grinding conditions (**b**).

## Data Availability

Data are contained within the article.

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
