# Peer review of "Selected Aspects of Precision Grinding Processes Optimization"

_materials, 2024, doi:10.3390/ma17030607_

Round 1
Reviewer 1 Report
Comments and Suggestions for Authors
Interesting dicussion. Minor improvements should be considered:
-In Section 4, only the last paragraph considers tool life. Maybe the influence of tool wear could be discussed regarding the application of the described models.
-The section numbering should be revised.
-In Figures 7 and 8, either a reference or more details on the grinding process and measurements should be provided.
Author Response
Dear Reviewer, thank you for considering our manuscript for publication in Materials. We are grateful for the valuable suggestions provided. In the revised manuscript, all changes are highlighted in red.
Below, we submit responses to the comments.
Comment, remark 1: In Section 4, only the last paragraph considers tool life. Maybe the influence of tool wear could be discussed regarding the application of the described models.
Response 1: In this section (after the change, it is now subsection 2.3) we described the correct approach to developing a model for optimizing the grinding process taking into account tool life. We discussed the correct methodology. The tool life model is described in the next section (after the numbering change, it is section no. 4). The description of this section has been expanded in line with your comment. The next section also highlights the application areas of the described models (as suggested).
Comment, remark 2: The section numbering should be revised.
Response 2: The section numbering has been corrected and the number of sections reduced from 7 to 5.
Comment, remark 3: In Figures 7 and 8, either a reference or more details on the grinding process and measurements should be provided.
Response 3: In Figure 7, we have added the values of the process parameters that were used in the script simulating the temperature field in the grinding process for different velocities of the heat source. In Figure 8, and in its discussion, was extended a detailed description of the methodology for its development.
The authors thank the Reviewer for their time spent analysing the manuscript and valuable comments.

Reviewer 2 Report
Comments and Suggestions for Authors
Dear Authors,
I extend sincere congratulations to you for the diligent work undertaken in crafting the paper titled "Selected Aspects of Precision Grinding Processes Optimization." Your selective approach to optimizing grinding processes, considering the probabilistic features inherent in this domain, is truly commendable. The research outlines significant aspects of the grinding process optimization, addressing the dispersion of quantities, uncertainties in research results, and limitations in information extraction procedures.
The attention given to uncertainties in research results, imperfections in information extraction, and the restricted availability of data while employing simulation and regression models in optimization procedures is acknowledged. Your analysis of determining the durability of abrasive tools in the optimization process, accounting for the dispersion of controlled process parameters, adds depth to the discussion. Furthermore, acknowledging the effects of interference in relationships describing grinding efficiency and costs is a valuable perspective in the optimization landscape.
Regarding the structure of the paper, I would like to suggest the following adaptations:
Abstract:
This section should include a concise introduction to the paper's subject, outline the research objectives, communicate the key message to the reader, detail the research methodology, and encapsulate the results and conclusions of the study.
Introduction:
Within this section, the current state of research should be described, emphasizing the necessity of the study by highlighting literature gaps. It should present a concise overview of the main research originality/novelty, articulate the primary research goals, and outline the paper's structure in a dedicated paragraph.
Introduce a section of Results and Discussions wherein the presented information in the paper will be restructured as follows:
Results:
This section should elucidate the occurrences during the study, detailing the discoveries or confirmations made. It should provide a simple description of the data, incorporate statistical analyses, evaluate observed trends, and explain the significance of results for a broader understanding, referencing relevant published research. A critical analysis of the collected data should also be presented.
Discussions:
In this section, any gaps or inconsistencies in the research should be addressed. Additionally, potential ways for future research to confirm conclusions or further advance the study should be proposed.
Conclusions:
The concluding section should encompass reflections on the research's purpose, whether achieved or not. It should highlight the specific contributions made in the paper, both theoretical and practical, and suggest directions for future research.
This structure will likely enhance the coherence and comprehensiveness of your work, enabling readers to navigate through the essential stages and contributions of your research.
Once again, congratulations to the authors for their commendable work and dedication in contributing to the advancement of knowledge in this field.
Best regards,
Reviewer
Comments on the Quality of English LanguageEnglish is fine.
Author Response
Dear Reviewer, thank you for review our manuscript for publication in Materials. We are grateful for the valuable suggestions provided. In the revised manuscript, all changes are highlighted in red.
Below, we submit responses to the comments to the review.
I extend sincere congratulations to you for the diligent work undertaken in crafting the paper titled "Selected Aspects of Precision Grinding Processes Optimization." Your selective approach to optimizing grinding processes, considering the probabilistic features inherent in this domain, is truly commendable. The research outlines significant aspects of the grinding process optimization, addressing the dispersion of quantities, uncertainties in research results, and limitations in information extraction procedures.
The attention given to uncertainties in research results, imperfections in information extraction, and the restricted availability of data while employing simulation and regression models in optimization procedures is acknowledged. Your analysis of determining the durability of abrasive tools in the optimization process, accounting for the dispersion of controlled process parameters, adds depth to the discussion. Furthermore, acknowledging the effects of interference in relationships describing grinding efficiency and costs is a valuable perspective in the optimization landscape.
Regarding the structure of the paper, I would like to suggest the following adaptations:
Abstract:
This section should include a concise introduction to the paper's subject, outline the research objectives, communicate the key message to the reader, detail the research methodology, and encapsulate the results and conclusions of the study.
Introduction:
Within this section, the current state of research should be described, emphasizing the necessity of the study by highlighting literature gaps. It should present a concise overview of the main research originality/novelty, articulate the primary research goals, and outline the paper's structure in a dedicated paragraph.
Introduce a section of Results and Discussions wherein the presented information in the paper will be restructured as follows:
Results:
This section should elucidate the occurrences during the study, detailing the discoveries or confirmations made. It should provide a simple description of the data, incorporate statistical analyses, evaluate observed trends, and explain the significance of results for a broader understanding, referencing relevant published research. A critical analysis of the collected data should also be presented.
Discussions:
In this section, any gaps or inconsistencies in the research should be addressed. Additionally, potential ways for future research to confirm conclusions or further advance the study should be proposed.
Conclusions:
The concluding section should encompass reflections on the research's purpose, whether achieved or not. It should highlight the specific contributions made in the paper, both theoretical and practical, and suggest directions for future research.
This structure will likely enhance the coherence and comprehensiveness of your work, enabling readers to navigate through the essential stages and contributions of your research.
Once again, congratulations to the authors for their commendable work and dedication in contributing to the advancement of knowledge in this field.
Response. Thank you very much for your positive opinion about our article. We have improved the structure of the article. The number of sections has been reduced from 7 to 5. In addition, as suggested, we have indicated which content from the subsections is the result of research results.
The authors thank the Reviewer for their time spent analysing the manuscript and valuable comments.

Reviewer 3 Report
Comments and Suggestions for Authors
The article uses a high number of literature sources to deeply investigate the selected topic of grinding process precision optimization. It is well-organized, well-written. The language is judged to be fine.
The paper details the factors that affect the grinding process itself, paying attention to the most relevant economical and technical factors. Four major groups are mentioned from the perspective of goals and criteria. It is revealed that the grinding force is influenced by the feed speed and cutting depth, while the temperature in the cutting zone mostly depends on the cutting speed and rake angle of the cutting edge. The paper picks some important factors and describe the optimization process of grinding, incorporating the causes for the uncertainty factors as well. It is demonstrated that with proper parameter selection (optimization), the life of the grinding wheel can be extended significantly.
Questions:
1. How did the authors get the results of Fig. 7? If it is taken from the literature, please indicate it. If it is based on a model, please describe the most important properties.
2. How did the authors get the results of Fig. 8? If it is taken from the literature, please indicate it. If it is the finding of their own model, please describe the modelling aspects.
Author Response
Dear Reviewer, thank you for considering our manuscript for publication in Materials. We are grateful for the valuable suggestions provided. In the revised manuscript, all changes are highlighted in red.
Below, we submit responses to the comments to the review.
The article uses a high number of literature sources to deeply investigate the selected topic of grinding process precision optimization. It is well-organized, well-written. The language is judged to be fine.
The paper details the factors that affect the grinding process itself, paying attention to the most relevant economical and technical factors. Four major groups are mentioned from the perspective of goals and criteria. It is revealed that the grinding force is influenced by the feed speed and cutting depth, while the temperature in the cutting zone mostly depends on the cutting speed and rake angle of the cutting edge. The paper picks some important factors and describe the optimization process of grinding, incorporating the causes for the uncertainty factors as well. It is demonstrated that with proper parameter selection (optimization), the life of the grinding wheel can be extended significantly.
Thank you for your positive opinion about our article.
Point 1: How did the authors get the results of Fig. 7? If it is taken from the literature, please indicate it. If it is based on a model, please describe the most important properties.
Response 1: The graphs (Figure 7) show the temperature field for grinding for alloy constructional steel for different speeds of the heat source (longitudinal grinding feed) 0.005, 0.01, 0.02, 0.04, 0.08, 0.16, 0.32, 0.64 m/s, for an initial temperature of 290 K, heat source efficiency depending on the feed speed and ranging from 20 to 130 W/mm2, for the density of the material 7820 kg/m3, specific heat 490 J/(kg∙K) and thermal conductivity 42 W/(m∙K). The length and width of the cooling zone were 40 and 20 mm, respectively. The displacement of the cooling zone relative to the center of the grinding zone in the direction opposite to the direction of the feed movement was 10 mm.
In line with the comment, the content of the section in the discussion of results has been modified.
The simulation results indicate beneficial thermal effects resulting from increasing the longitudinal feed speed. To ensure the required surface roughness for a given surface machining efficiency, it is possible to reduce the value of the transverse feed.
Point 2: How did the authors get the results of Fig. 8? If it is taken from the literature, please indicate it. If it is the finding of their own model, please describe the modelling aspects.
Response 2: Figure 8 illustrates the impact of constraints imposed on the features characterizing the state and results of the machining process. The impact of two strategies on the decision-making process regarding the end of the grinding wheel's life and the required renewal of its working surface was compared.
The first strategy consisted in pre-emptive renewal of the grinding wheel surface without supervision, but only using the results of previous diagnostic procedures. The second strategy used continuous monitoring of the process state based on the same characteristics.
The following features were adopted to describe the process state: c1 - normalized value of the grinding force in N, c2 - normalized value of the Sp5 parameter of the height of surface irregularities in µm, c3 - normalized value of the deviation of the diameter of the grinded cylindrical surface in µm.
Normalization of the values of these parameters to the range <0, 1> allowed for visualization of the problem and comparison of data in a common range. In the normalization performed, 0 corresponds to the lowest value of a given feature in the analyzed process, and 1 corresponds to the highest value that could occur during the durability of the grinding wheel.
It was taken into account that the durability distributions determined for each feature separately will, of course, differ in expected value and variance. The form of these distributions, as a result of the mainly multiplicative mechanism of accumulation of the influence of factors influencing changes in individual features, will be similar to the log-normal distribution.
The durability of the grinding wheel is the shortest of the periods during which individual controlled features reach or would reach the limit permissible value.
The above observations show that in machining conditions in which the condition of the tool and process characteristics are monitored, in subsequent periods of use of the grinding wheel, various limiting conditions may determine the durability periods, but the average durability will be at a level much higher than that corresponding to pre-emptive refurbishment of the active surface of the tool.
In line with the comment, the content of the section in the discussion of results has been modified.
The authors thank the Reviewer for their time spent analysing the manuscript and valuable comments.
